| Virology | Research Article

# Analysis of the genetic evolution and recombination of the PRRSV-2 GP2 protein in China from 1996 to 2023

Kexin Liu,[1] Chen Lv,[1] Cuihua He,[1] Jiankun Pang,[1] Chunyao Lai,[1] Siliang Chen,[1] Ruining Wang,[2] Weili Kong,[3] Jun Ma,[1] Mengmeng Zhao[1]

**ABSTRACT** Porcine reproductive and respiratory syndrome (PRRS) is among the most serious infectious diseases of pigs worldwide. It is caused by PRRSV and frequently mutates and recombines. To date, however, there have been relatively few studies that have analyzed the GP2 membrane protein of this virus. In this study, we compared 570 nucleotide sequences of the PRRSV-2 GP2 protein obtained from the NCBI GenBank database, which were subjected to phylogenetic analysis. We selected 64 representative strains to investigate the genetic evolution and recombination of the GP2 protein in China. Lineages 1 and 8 were the most prevalent, while lineages 5 and 8 showed closer genetic relationships. The nucleotide similarities of the 570 GP2 sequences ranged from 83.0% to 100%, with amino acid similarities from 80.2% to 100%. Recombinant analysis indicated lineage 1 strains had the highest recombination probability. Comparison of amino acid sequences showed substitutions without deletions or insertions, with lineage 1 exhibiting the most substitutions and lineage 8 the fewest. These findings enhance understanding of PRRSV-2 genetic variation and provide a foundation for further studies on GP2 and vaccine development.

**IMPORTANCE** Porcine reproductive and respiratory syndrome virus (PRRSV) has caused significant losses and posed threats to the swine breeding industry. To date, there have been comparatively few studies on the GP2 protein of PRRSV-2, and consequently, many unanswered questions remain regarding its pathogenicity-associated mechanisms and effects. We collected 570 nucleotide sequences of the GP2 protein of this virus and used these data to perform multifaceted analytical work in order to facilitate the understanding of the genetic evolution of this virus and recombinant mutations. These provide basic data for the follow-up study of GP2 and lay a foundation for further in-depth studies of this virus and vaccine development.

**KEYWORDS** porcine reproductive and respiratory syndrome virus, GP2 protein, phylogeny, genetic variation, recombination analysis

Porcine reproductive and respiratory syndrome (PRRS), caused by the PRRS virus (PRRSV), is an acute infectious disease characterized by reproductive disorders, including abortions, weak fetuses, and mummified fetuses in gestating sows. Moreover, it causes respiratory symptoms and high mortalities in pigs of all ages (1) and has accordingly resulted in considerable economic losses to the pig industry, both in China and worldwide. The emergence of PRRS was first reported in the United States in 1987 (2), and subsequently, there has been a succession of large-scale PRRSV infections in pigs in some European countries. PRRSV was first detected in Taiwan, China, in 1991 (3), and in 1996, researchers at the Harbin Veterinary Research Institute of the Chinese Academy of Agricultural Sciences isolated PRRSV for the first time from a suspected case of PRRS in China, which they named CH-1a (4). PRRSV has now been endemic in China for nearly 30

**Peer Reviewer** Jayeshbhai Mansinhbhai Chaudhari, University of Nebraska-Lincoln, Lincoln, Nebraska, USA

Address correspondence to Mengmeng Zhao, mengmengzhao2021@fosu.edu.cn, or Jun Ma, majun@fosu.edu.cn.

The authors declare no conflict of interest.

See the funding table on p. 13.

years (5), over which time it has been constantly re-organizing and mutating during the course of epidemics (6–8).

PRRSV can be categorized into two genotypes, namely PRRSV-1 and PRRSV-2, the latter of which is predominantly endemic in China, including both CH-1a and BJ-4, which were the first strains isolated in China (9). In 2006, China saw the emergence of highly pathogenic PRRSV (HP-PRRSV) (10, 11), characterized by a discontinuous deletion of 30 amino acids in the sequence of the NSP2 protein (12), which is believed to have evolved from the continuous mutation of a CH-1a-like PRRSV (13), with the representative strain being JXA1.

PRRSV is a single-stranded, positive-stranded RNA virus of the family *Arteriviridae* under the order *Nidoviridae* (6, 14). The viral particles are spherical, typically 60–65 nm in diameter (15), with capsular membrane and icosahedral symmetry. The single-stranded PRRSV genome is approximately 15 kb in total length, contains a 5′-terminal cap and a 3′-terminal poly(A) tail, and comprises at least 10 open reading frames (ORFs). Among these ORFs, ORF2a encodes the GP2 protein (sometimes referred to as GP2a), a type I integral membrane protein (16). Similarly, ORF3, ORF4, and ORF5 encode the GP3, GP4, and GP5 proteins, respectively. In addition, NSP4, a nonstructural protein, functions to promote viral replication (17). GP2 has a molecular weight of approximately 29–30 kDa, a total length of 771 bp, and two predicted N-glycosylation sites (18).

GP4, GP5, and M proteins interact with each other (19), as do GP2, GP3, and GP4 as minor vesicle membrane proteins. These latter three proteins can form covalent bonds, thereby forming a heterotrimer, which is important for the composition of infectious virus particles (20, 21). GP2 has been established to induce the proliferation of T lymphocytes in hosts (22), along with the production of specific neutralizing antibodies against PRRSV. In this regard, it has been found that the partial deletion of the amino acid at the carboxyl terminus of GP2 has no significant effects on the growth or replication of the recombinant virus in Marc-145 cells (23), and does not alter the neutralizing antigen of this recombinant virus. Furthermore, GP2 has been shown to bind to the CD163 receptor on porcine alveolar macrophages, and blocking this binding process has been demonstrated to reduce the entry of PRRSVs into host cells (24–26). Consequently, it is reasonably speculated that the GP2 protein influences the host cell invasion of PRRSV and also modulates the host immune response. Indeed, Welch et al. (27) have found that viruses lacking the ORF2 and ORF4 genes are unable to survive on Marc-145 cells and porcine alveolar macrophages, thereby indicating that the GP2 and GP4 proteins are essential for viral replication. It has also been shown that GP2-activated transcription factors may activate viral anti-apoptotic mechanisms and inhibit the expression of viral pro-apoptotic genes (28, 29). In summary, the GP2 protein plays a key role in viral invasion, replication, and apoptosis and is closely related to the specific antibodies produced by the host immune response. Therefore, a detailed evolutionary analysis of GP2 would be beneficial in discovering ways to effectively block the invasion of PRRSV into host cells.

To date, there have been comparatively few studies on the GP2 protein of PRRSV-2, and consequently, there remain many unanswered questions regarding its pathogenicity-associated mechanisms and effects. In this study, using 570 nucleotide sequences of PRRSV-2 GP2, we examined the genetic evolution and recombination of this protein and constructed phylogenetic trees. Among these sequences, we analyzed the amino acid and nucleotide similarities of 64 representative sequences. Using these analyses, we also sought to identify recombination events. The findings of this study will enable us to gain a better understanding of the prevalence and evolution of PRRSV in China and provide insights regarding the genetic variation of the GP2 protein, which will thereby provide a basis for the development of PRRSV-2-related drugs, vaccines, and subsequent therapeutic treatments.

## MATERIALS AND METHODS

### PRRSV-2 GP2 sequence data collection

The 570 reference sequences of the GP2 protein of PRRSV-2 strains analyzed in this study were obtained from the GenBank database on the NCBI website. Of these, 567 sequences were from China and the remaining 3 were from the United States. These sequences are derived from strains in lineages 1, 3, 5, and 8 isolated over a span of 32 years, including classical and representative strains of each lineage (30). This broad coverage was designed to provide a comprehensive view of the composition and variation of the GP2 protein.

### GP2 sequence alignment

All nucleotide sequences were compared using the MegAlign program in the DNASTAR package (version 7.0, Madison, WI, USA). Using this program, we performed Clustal W analysis of the similarities between sequences. These sequences were subsequently translated using the Editseq program in the DNASTAR package to yield the corresponding amino acid sequences of GP2 in the 570 PRRSV-2 strains. Among these amino acid sequences, 64 PRRSV GP2 sequences were selected (Table 1). The criteria for sequence selection included classical strains, strains often used as lineage representatives in the literature, vaccine strains commonly used in China, and newly emerged strains in recent years. The classical strains included VR2332, RespPRRS MLV, and NADC34 from the United States, and CH-1a and BJ-4 from China. Lineage representative strains include HENAN-XINX (lineage 1), JL580 (lineage 1), QYYZ (lineage 3), FJFS (lineage 3), S1 (lineage 5), GS2003 (lineage 5), JXA1 (lineage 8), HUN4 (lineage 8), and so on. These sequences were compared and analyzed for mutation sites using the Clustal W method in MegAlign. The Hiplot website (https://hiplot.com.cn/) was used to create a heatmap.

### Phylogenetic analysis

Phylogenetic trees were constructed using the neighbor-joining (NJ) and maximum likelihood (ML) methods in MEGA software (version 7.0; Center for Evolution and Informatics, Tempe, AZ, USA). Using both methods, we performed 1,000 bootstrap replications to generate the respective phylogenetic trees, with the remaining parameters being left unchanged. Subsequently, the generated phylogenetic trees were embellished and annotated using the Interactive Tree of Life (embl.de) (iTOL) website.

### Recombination analysis

To predict recombination events for the 570 nucleotide sequences, we used RDP (Version 4.0) software based on seven reference methods for recombination analysis, namely, RDP, BootScan, GENECONV, Chimaera, MaxChi, SiScan, and 3Seq. The RDP software was widely used for the detection of recombination events (31), allowing highly automated analysis of a large number of sequences with multiple methods (32). Recombination events that occur four or more times are considered standard recombination events, which were confirmed using validation methods. In addition, SimPlot (Version 3.5.1) was used to validate the standard reorganization events.

## RESULTS

### Nucleotide similarity

Sixty-four strains were selected from 570 strains as representatives for similarity comparison, which included classical and common strains from various lineages to ensure that the results obtained from the study were representative and meaningful. The similarity of the 64 GP2 nucleotide sequences ranged from 83.0% to 100.0%, and the lowest similarity between nucleotide sequences was between 15JX 1-2015 of lineage 1 and FJFS-2015 of lineage 3 with 83.0% similarity. Sequences with 100% similarity were

TABLE 1 The GP2 reference sequences of 64 PRRS strains

| Year | Area | Strain | GenBank accession number |
| --- | --- | --- | --- |
| 1992 | USA | VR2332 | EF536003 |
| 1996 | China | CH-1a | AY032626 |
| 1996 | China | BJ-4 | AF331831 |
| 1998 | USA | RespPRRS MLV | AF066183 |
| 2002 | China | HB-1-3.9c | HQ233605 |
| 2002 | China | GS2002 | EU880441 |
| 2003 | China | GS2003 | EU880442 |
| 2003 | China | HK1 | KF287132 |
| 2006 | China | CC-1 | EF153486 |
| 2006 | China | HEB1 | EF112447 |
| 2006 | China | HUB2 | EF112446 |
| 2006 | China | HUN4 | EF635006 |
| 2006 | China | JXA1 | EF112445 |
| 2006 | China | JXwn06 | EF641008 |
| 2006 | China | S1 | DQ459471 |
| 2006 | China | TJ | EU860248 |
| 2007 | China | 07HEN | FJ393457 |
| 2007 | China | GD | EU109503 |
| 2007 | China | GDQJ | GQ374441 |
| 2007 | China | HuN | EF517962 |
| 2007 | China | rV63 | EU360129 |
| 2007 | China | Shaanxi-2 | HQ401282 |
| 2008 | China | CH-1R | EU807840 |
| 2009 | China | APRRS | GQ330474 |
| 2009 | China | SD-09 | HQ843180 |
| 2009 | China | SD1-100 | GQ914997 |
| 2009 | China | SX-1 | GQ857656 |
| 2010 | China | 10-LW7-1 | JQ663567 |
| 2010 | China | GX1003 | JX912249 |
| 2010 | China | JX | JX317649 |
| 2010 | China | QY2010 | JQ743666 |
| 2011 | China | GM2 | JN662424 |
| 2011 | China | QYYZ | JQ308798 |
| 2011 | China | WUH4 | JQ326271 |
| 2012 | China | HENAN-HEB | JQ326271 |
| 2012 | China | JL-04-12 | JX177644 |
| 2013 | China | BJ-F60 | KP890339 |
| 2013 | China | HENAN-XINX | KF611905 |
| 2013 | China | JL580 | KR706343 |
| 2014 | China | CHsx1401 | KP861625 |
| 2014 | China | HENXX-1 | KU950372 |
| 2014 | China | JSWA | KY373214 |
| 2014 | USA | NADC34 | MF326985 |
| 2015 | China | 15GD1 | KX815407 |
| 2015 | China | 15HEB1 | KX815407 |
| 2015 | China | 15JX1 | KX815419 |
| 2015 | China | FJFS | KP998476 |
| 2015 | China | GZgy15-1 | KT358728 |
| 2015 | China | HNyc15 | KT945018 |

QYYZ-2011 and QY2010-2010, BJ-4-1996 and rV63-2007, JXwn06-2006 and HuN-2007, JXwn06-2006 and HEB1-2006, JXwn06-2006 and GDQJ-2007, Resp PRRS MLV and BJ-4, Resp PRRS MLV and rV63-2007, and HM1801-2018 and CN-H4-2018. Among these, the

similarities within HuN-2007, HEB1-2006, and GDQJ-2007 were all 100%. The similarities of nucleotide sequences within lineage 1 ranged from 83.9% to 100.0%, and the similarities of nucleotide sequences within lineage 8 ranged from 94.4% to 100.0%, which were the most and least differentiated lineages in terms of nucleotide similarity among the four lineages, respectively (Table 2).

## Amino acid similarity

The similarity of the 64 GP2 amino acid sequences ranged from 80.2% to 100.0%, with the lowest similarity of 80.2% between FJFS-2015 from lineage 3 and HLJTZJ864-2010-2020, FJ0908-2018, and CH-SCLS-2-2020 from lineage 1. Nucleotide sequences with a similarity of 100% have a corresponding amino acid sequence similarity of 100%. In addition to this, there were 15GD1-2015, CN-H4-2018, and HM1801-2018 in lineage 8, which had a similarity of 100% to each other, as well as the five strains GDQJ-2007, GD-2007, WUH4-2011, JXA1-2006, and HUB2-2006 in lineage 8, which were compared to each other and had a similarity of 100%. The similarity of amino acid sequences within lineage 1 ranged from 81.7% to 100.0%, and the similarity of amino acid sequences within lineage 8 ranged from 94.2% to 100.0%, which were the most and least different amino acid similarity among the four lineages, respectively (Table 2).

## Phylogenetic analysis

Based on the global classification system of PRRSV-2 viruses, the 570 assessed strains were categorized into lineages 1, 3, 5, and 8, and as indicated in the phylogenetic trees shown in Fig. 1 and 2, lineages 1 and 8 are currently the two major lineages prevalent in China. The strains isolated in 2023 are mainly from lineages 1, 5, and 8.

## Amino acid sequence comparisons

The amino acid sequences of the GP2 protein of the 64 representative strains were compared using the Clustal W method in MegAlign, which revealed that this protein consists of 256 amino acids with amino acid substitutions, although no insertions or deletions. At sites 1–129 of the amino acid sequence, lineage 1 commonly shows amino acid substitutions at sites 7, 9, 13, 19, 24, 26, 39, 45, 49, 78, 83, 88, 91, 98, and 118, whereas in lineage 3 strains, substitutions typically occur at sites 5, 13, 19, 42, 45, 49, 59, 78, 83, 84, 98, and 120. Slightly fewer substitutions were identified in lineage 5 sequences, at positions 5, 9, 10, 24, 32, 42, 91, and 122, and only three substitutions were detected in lineage 8 sequences at sites 23, 50, and 97 (Fig. 3).

At sites 130–256 of the amino acid sequence, common amino acid substitutions in lineage 1 sequences were found at sites 141, 174, 182, 187, 188, 189, 235, 237, 240, 245, 252, and 254, and at sites 174, 188, 189, 235, 237, 245, 251, 252, and 254 in lineage 3 strains. Lineage 5 amino acid substitutions are commonly found at sites 141, 174, 188, 235, 237, 240, and 256, whereas those in lineage 8 strains occur only at sites 250 and

**TABLE 2** Nucleotide (nt) and amino acid (aa) similarity analysis based on *GP2* gene (%)

|  |  | Lineage 1 | Lineage 3 | Lineage 5 | Lineage 8 |
|---|---|---|---|---|---|
| Lineage 1 | nt aa | 83.9–100.0[a] | 83.0–90.4[b] | 86.0–89.9 | 84.8–91.9 |
|  |  | 81.7–100.0[a] | 80.2–91.4[b] | 84.0–90.3 | 83.7–91.4 |
| Lineage 3 | nt aa |  | 89.4–100[a] | 87.7–91.1 | 87.4–91.1 |
|  |  |  | 87.2–100[a] | 85.6–91.1 | 86.0–92.6 |
| Lineage 5 | nt aa |  |  | 93.5–100.0[a] | 91.2–95.2 |
|  |  |  |  | 92.2–100.0[a] | 90.3–94.9 |
| Lineage 8 | nt aa |  |  |  | 94.4–100.0[a] |
|  |  |  |  |  | 94.2–100.0[a] |

[a]Indicates the highest homology.
[b]Indicates the lowest homology.

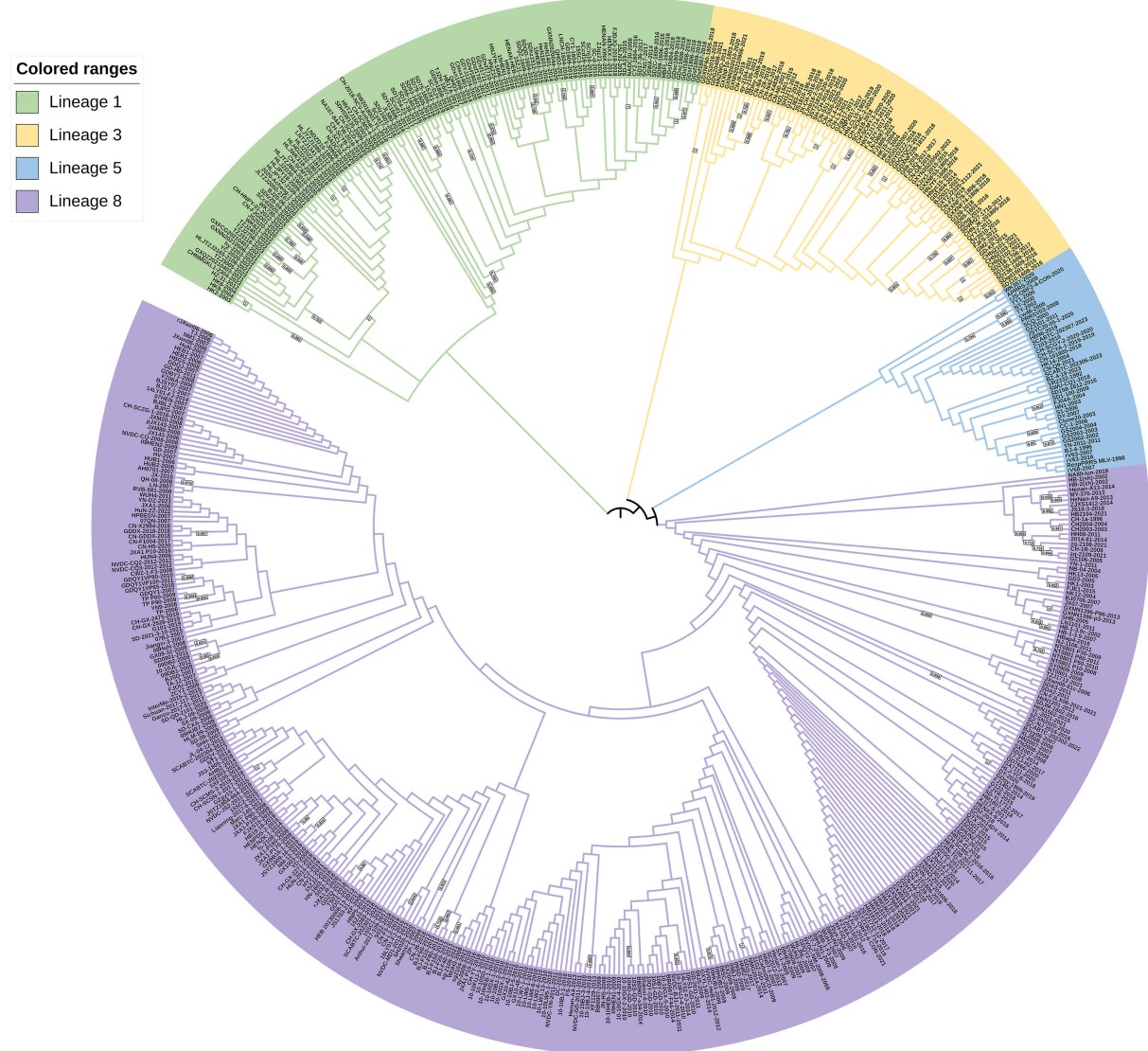

**FIG 1** Phylogenetic analysis based on sequences of the *GP2* gene. The phylogenetic tree was constructed using the maximum likelihood method with MEGA software, performed using 1,000 bootstrap replicates. The numbers on the phylogenetic tree branches are bootstrap values, with larger values representing higher branch confidence.

253. A heatmap of the different lineages with their common amino acid mutation sites is shown in Fig. 4. A better visualization of the differences in amino acid substitutions between lineages was shown.

Among the identified mutations, those specific to lineage 1 are T11→I11, T88→S88, and Y182→K182, although a few strains have other mutations, namely, the T88→V88 mutation in NADC34, HLJTZJ864, FJ0908, and CH-SCLS-2, and Y182→E182 in CHsx1401 and HENAN-HEB. The mutation sites specific to lineage 3 are Y59→F59 and E120→D120. Although sites that have undergone mutation in lineage 8 are not identical to the common mutation sites in other lineages, there are individual strains from other lineages that have mutations at the same sites as those identified in lineage 8. Consequently, lineage 8 strains do not carry any lineage-specific mutations.

In general, the GP2 amino acid sequence is characterized by a high frequency of amino acid substitutions at sites 42, 174, 188, 235, and 237, although lineage 8 strains invariably have mutations only at amino acid sites 23, 50, 97, 250, and 253.

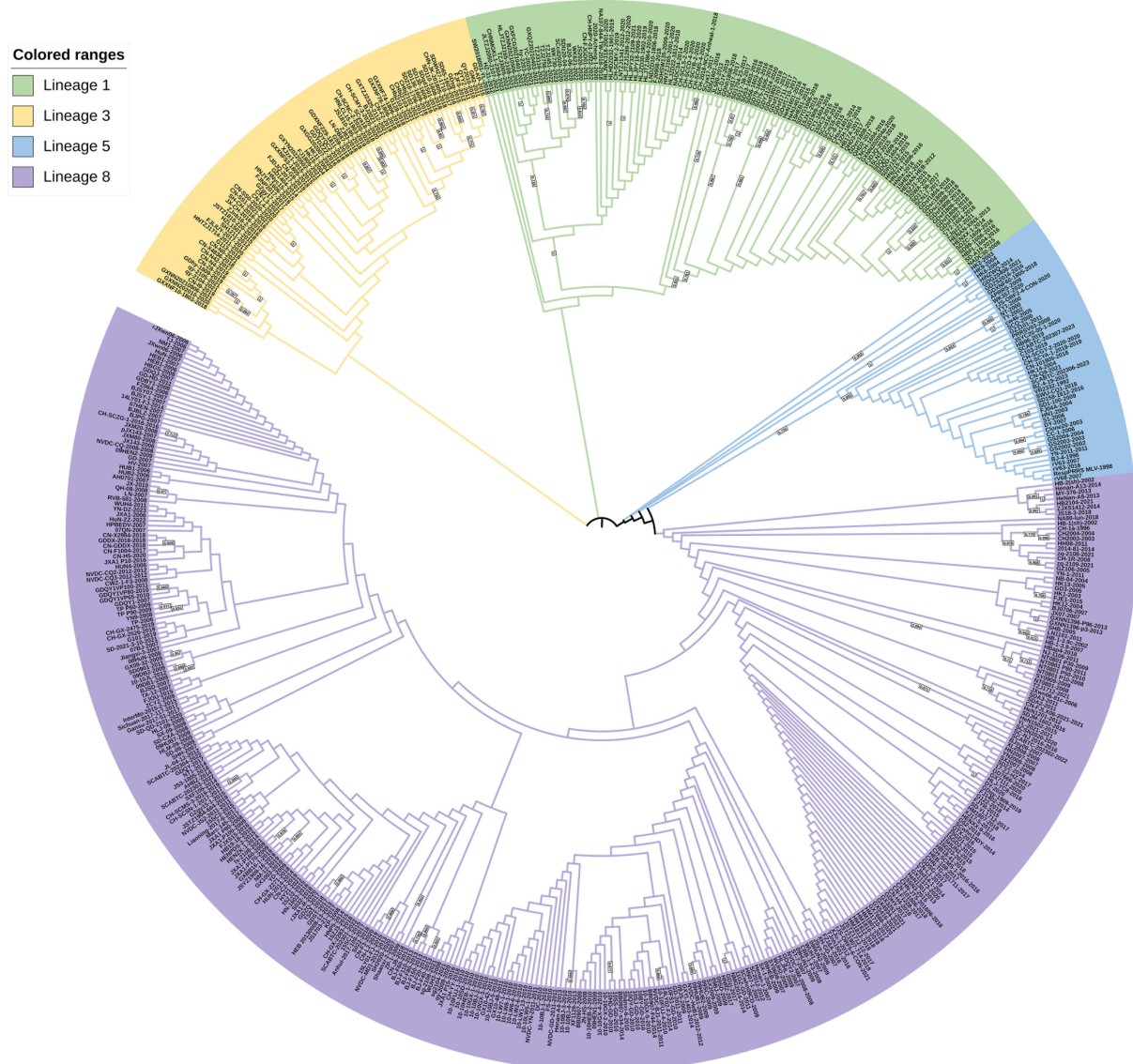

**FIG 2** Phylogenetic analysis based on sequences of the *GP2* gene. The phylogenetic tree was constructed using the neighbor-joining method with MEGA software, performed using 1,000 bootstrap replicates. The numbers on the phylogenetic tree branches are bootstrap values, with larger values representing higher branch confidence.

## Recombination analysis

The CH-1a strain was the earliest PRRSV strain isolated in China, and thus, we used the GP2 sequence of this strain as a control for the prediction of recombination events using RDP software (Fig. 5). We accordingly identified two events that met the criteria for recombination (Table 3), which were validated using SimPlot (Fig. 6). The two strains in which recombination was detected are both lineage 1 strains.

Recombination event 1 was the recombination of SCN17-2017 from lineage 1 and GXXNF53-1805-2018 from lineage 3 as the main and minor parental strains, respectively, to obtain SCya17-2017 from lineage 1. Recombination event 2 was the recombinant combination of XF1129-2013 from lineage 8 as the main parent strain and SC-d-2015 from lineage 1 to obtain JL580-2013 from lineage 1. The *P* value for both reorganization events was less than 0.05, which is statistically significant.

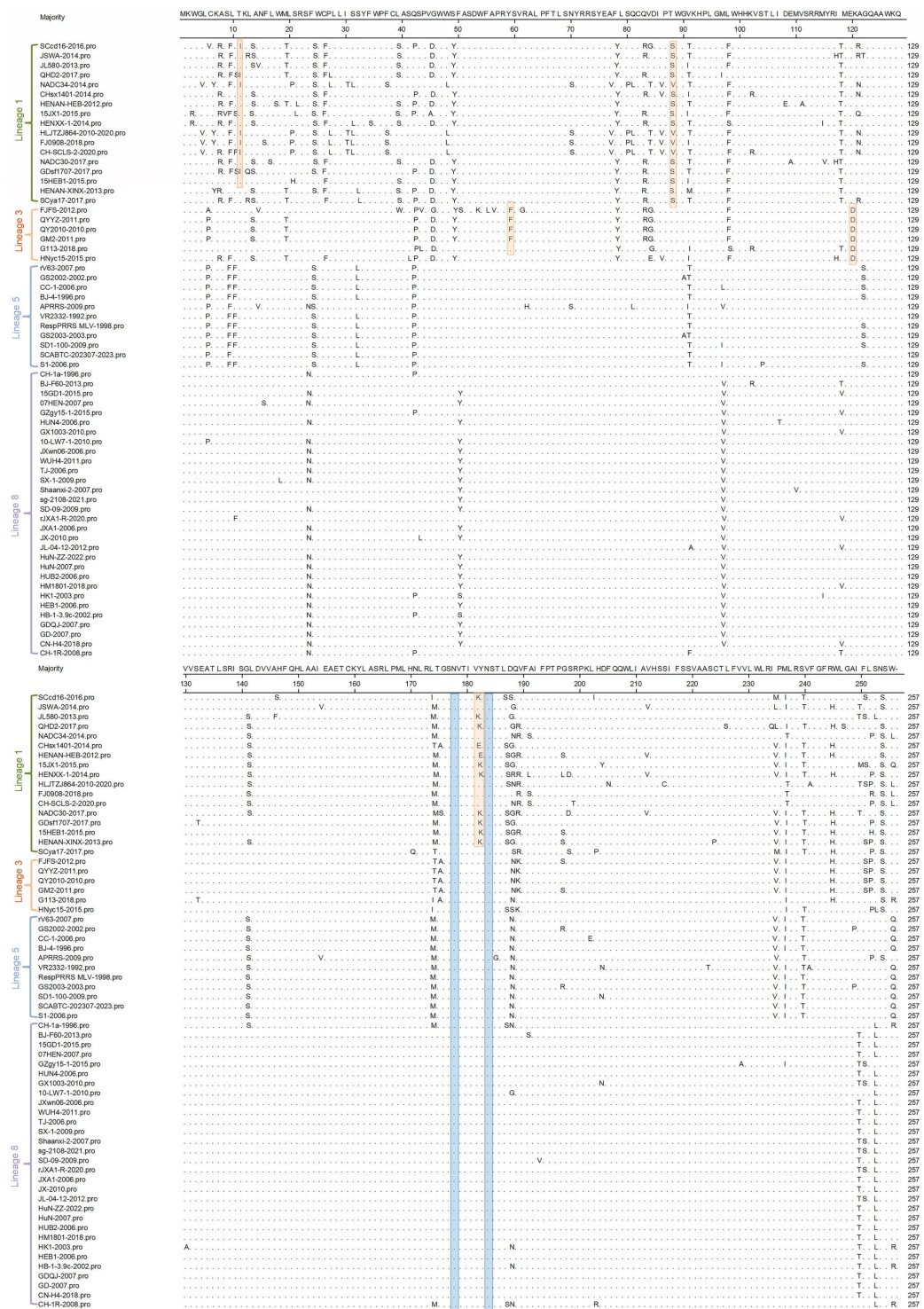

**FIG 3** Alignment of 64 PRRSV-2 GP2 amino acid sequences from lineages 1, 3, 5, and 8 strains. The orange regions represent lineage-specific mutations. The blue regions represent glycosylation sites.

## DISCUSSION

PRRSV has been endemic in mainland China since it was detected in 1995, with recombination and mutation occurring continuously. Five hundred seventy PRRSV-2 GP2 sequences were selected, including classical strains, highly pathogenic strains, and representative strains of each lineage. Three of the US strains were VR-2332, NADC34,

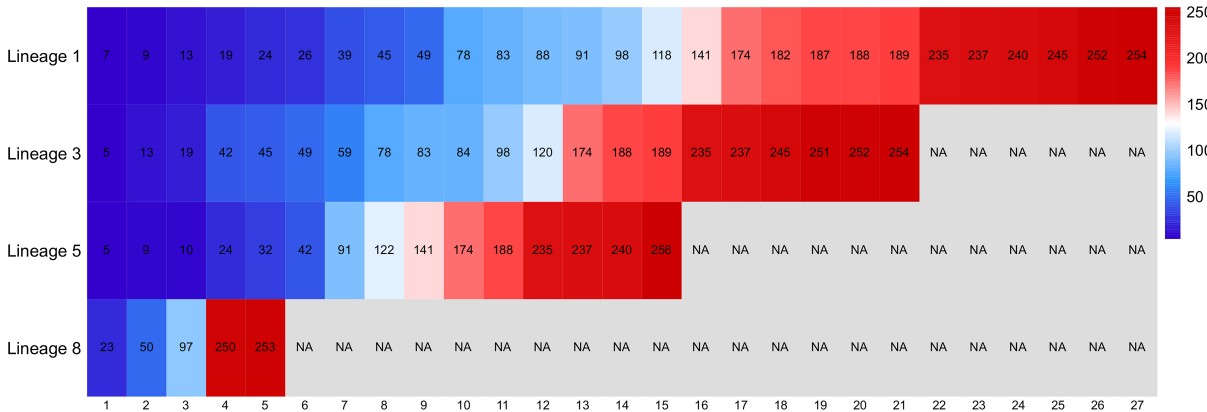

**FIG 4** Heatmap of different lineages with their common amino acid mutation sites. The values in the legend represent the sites of amino acid mutations. The numbers at the bottom show how many common mutation sites there are in each lineage.

and Resp PRRS MLV. Both VR-2332 and Resp PRRS MLV belonged to lineage 5, and both showed the highest similarity to BJ-4, with 99.2% versus 100% similarity in nucleotide sequences and 98.4% versus 100% similarity in amino acid sequences, respectively. NADC34 belongs to lineage 1 and has the highest similarity to CH-SCLS-2 with 99.4% nucleotide sequence similarity and 98.1% amino acid sequence similarity. The main prevalent strains in China are still lineages 1 and 8 strains, and the strains in 2023 are also mostly from lineages 1, 5, and 8. Half of the parental strains of the two recombination events belonged to lineage 1, and all of the recombinant strains belonged to lineage 1. Lineage 1 strains are likely to undergo frequent recombination and generate new strains as they circulate in China, resulting in an increase in the number of new lineage 1 strains and an accelerated rate of circulation. SCya17 and JL580, as recombinant strains, differed significantly in pathogenicity, with JL580 being a highly pathogenic strain and SCya17 showing low pathogenicity in clinical investigations (33). Recombination has resulted in significant genetic diversity of NADC30-like PRRSV in China (34), which is one of the reasons for the large differences in viral pathogenicity. The results of amino acid sequence comparisons surface that amino acid mutations occur most frequently in lineage 1 strains, which should be related to the large number of recombinant mutations that have occurred. There were three mutation sites specific to lineage 1, which is consistent with the generally lower similarity between lineage 1 and the other lineages, and the greatest differences in similarity among the internal strains.

The nucleotide sequence similarity of the 64 GP2s ranged from 83.0% to 100.0%, and the amino acid sequence similarity ranged from 80.2% to 100.0%. The minimum values of nucleotide and amino acid similarity of GP3 and GP5 are lower than the minimum values of nucleotide and amino acid similarity of GP2 (35, 36). Compared with GP3 and GP5, which are also structural proteins, GP2 is more conserved and less heterogeneous. Since 2006, HP-PRRSV broke out in southern China and spread rapidly throughout the country. HP-PRRSV is a mutation of CH-1a-like in the course of the epidemic in China, and HP-PRRSV belongs to lineage 8. It is widely accepted that lineage 8 is characterized by high recombination frequency. For example, in China, recombination events of NSP2, the nonstructural protein of PRRSV-2, occurred mainly in lineage 8 (37). However, according to the recombination and amino acid mutation of GP2 in this study, lineage 1 is more consistent with the characteristics of high recombination frequency and multiple mutations. Lineage 1 has significantly more amino acid mutation sites than the other lineages and has the largest difference in similarity, with the similarity of nucleotide sequence ranging from 83.9% to 100%, and the similarity of amino acid ranging from 81.7% to 100%. In contrast, lineage 8 strains had the least number of amino acid substitutions, with only five common mutation sites, and had the most conserved amino acid sequences of all the lineages. Lineage 8, which showed 94.4–

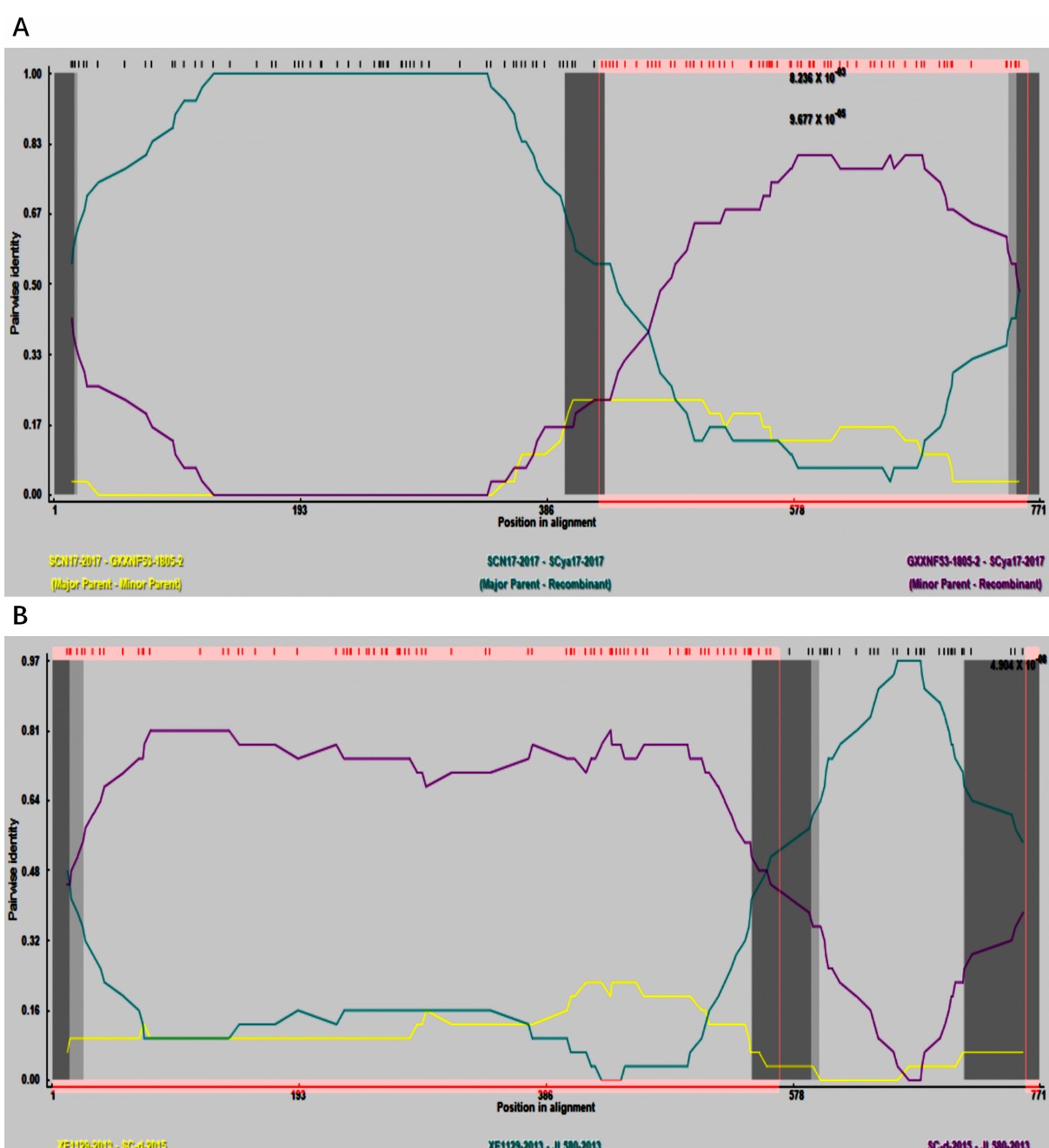

**FIG 5** Prediction of *GP2* gene recombination events by RDP. (A) Recombination analysis results of the recombinant strain SCya17-2017. (B) Recombination analysis results of the recombinant strain JL580-2013. The green line represents the homology between the major parental strain and the recombinant strain. The purple line represents the homology between the minor parental strain and the recombinant strain. The yellow line represents the homology between the major and minor parental strains.

100% nucleotide sequence similarity and 94.2–100% amino acid sequence similarity, was involved in only one standard recombination event as a minor parental strain. Lineage 8 tends to be conservative, indicating that this lineage is not as popular as lineage 1 in China in recent years, and the strains of lineage 1 have better epidemic conditions in China. Conventional vaccines for PRRSV in the Chinese market are mainly developed with lineage 8 strains, such as CH-1R, GDr180, JXA1-R, and TJM-F92. As a result, the cross-protection of conventional vaccines against lineage 1 is weak, which makes lineage

**TABLE 3** Recombination analysis of *GP2* gene

| Recombination event | Recombinant strian (lineage) | Main parental strain (lineage) | Minor parental strain (lineage) | Recombinant breakpoint | Recombination analysis method |
|---|---|---|---|---|---|
| 1 | SCya17-2017 (1) | SCN17-2017 (1) | GXXNF53-1805-2018 (3) | 400–431 (747-19) | RDP ($P = 1.161 \times 10^{-4}$) GENECONV ($P = 4.296 \times 10^{-2}$) BootScan ($P =$ NS) MaxChi ($P = 5.481 \times 10^{-9}$) Chimaera ($P = 2.230 \times 10^{-9}$) SiScan ($P = 1.043 \times 10^{-6}$) 3Seq ($P = 6.449 \times 10^{-14}$) |
| 2 | JL580-2013 (1) | XF1129-2013 (8) | SC-d-2015 (1) | 712-25 (546–599) | RDP ($P = 5.888 \times 10^{-8}$) GENECNV ($P =$ NS) BootScan ($P =$ NS) MaxChi ($P = 7.895 \times 10^{-5}$) Chimaera ($P =$ NS) SiScan ($P = 6.868 \times 10^{-11}$) 3Seq ($P = 5.881 \times 10^{-9}$) |

1 more susceptible to widespread epidemics and recombination. Therefore, vaccine development should consider strengthening the prevention against the more prevalent lineage strains in recent years. Compared to the GP5 protein of lineage 8, the GP5 protein was recombined at a higher frequency with greater differences in similarity. In connection with the idea that GP2 proteins influence the invasion of virulent strains into host cells, the high conservation of lineage 8 in the GP2 may correlate with the high pathogenicity of lineage 8 strains (38). It is hypothesized that stable expression of the GP2 protein contributes to the enhancement of lineage 8's ability to infect host cells and increase the impact of the strain on the host immune response.

In Taiwan, an outbreak of PRRS occurred as early as 1991, and the earliest report of PRRS in mainland China was in 1995, with the isolation of the PRRS virus by researchers in 1996. Six strains from Taiwan were selected for comparison of similarity with the reference strains, and it was found that lineage 5 had the highest similarity with the strains from Taiwan, with nucleotide similarity of 91.5–94.4% and amino acid similarity of 90.2–94.6%. The nucleotide similarity between the remaining three lineages and the Taiwan strains ranged from 85.5% to 91.6%, and the amino acid similarity ranged from 83.2% to 91.4%. This indicated that the possibility of recombination between lineage 5 strains from mainland China and Taiwan strains was extremely high, suggesting that the possibility of transmission of Taiwan strains to mainland China was high.

The glycosylation sites of GP2 are 178 and 184, and glycosylation at position 184 has been shown to be essential for PRRSV transmission (39). Amino acid sequence analysis showed no mutations in either glycosylation site, indicating that the glycosylation sites used for sequence analysis of the strains were highly conserved. The T98→S98 mutation decreases the infectivity of PRRSV-2 toward Marc-145 cells (40). This mutation is known to occur in lineages 1 and 3 based on amino acid sequence comparison, which suggests that lineages 1 and 3 may be less effective at isolating the virulent strain when Marc-145 cells are used. According to the results of the amino acid sequence comparison, no amino acid substitution occurred at K160, and the degree of conservation was extremely high. When K160 was replaced, the infectivity of PRRSV to PAMs was significantly reduced, and the replication of Low-Passage PRRSV Strain in pigs was reduced (41), which can be applied to the experimental production of vaccines to improve vaccine safety. GP2, GP3, and GP4 interact and form heterotrimers by forming covalent bonds to enter PRRSV particles. Therefore, the study of the glycosylation site of GP2 and nearby fragments may be useful to achieve the effect of blocking the composition of heterotrimers and thus affecting the infectivity of the virus. If the infectivity of PRRSV is made to decrease significantly, it will be very beneficial to control the spread of the disease and

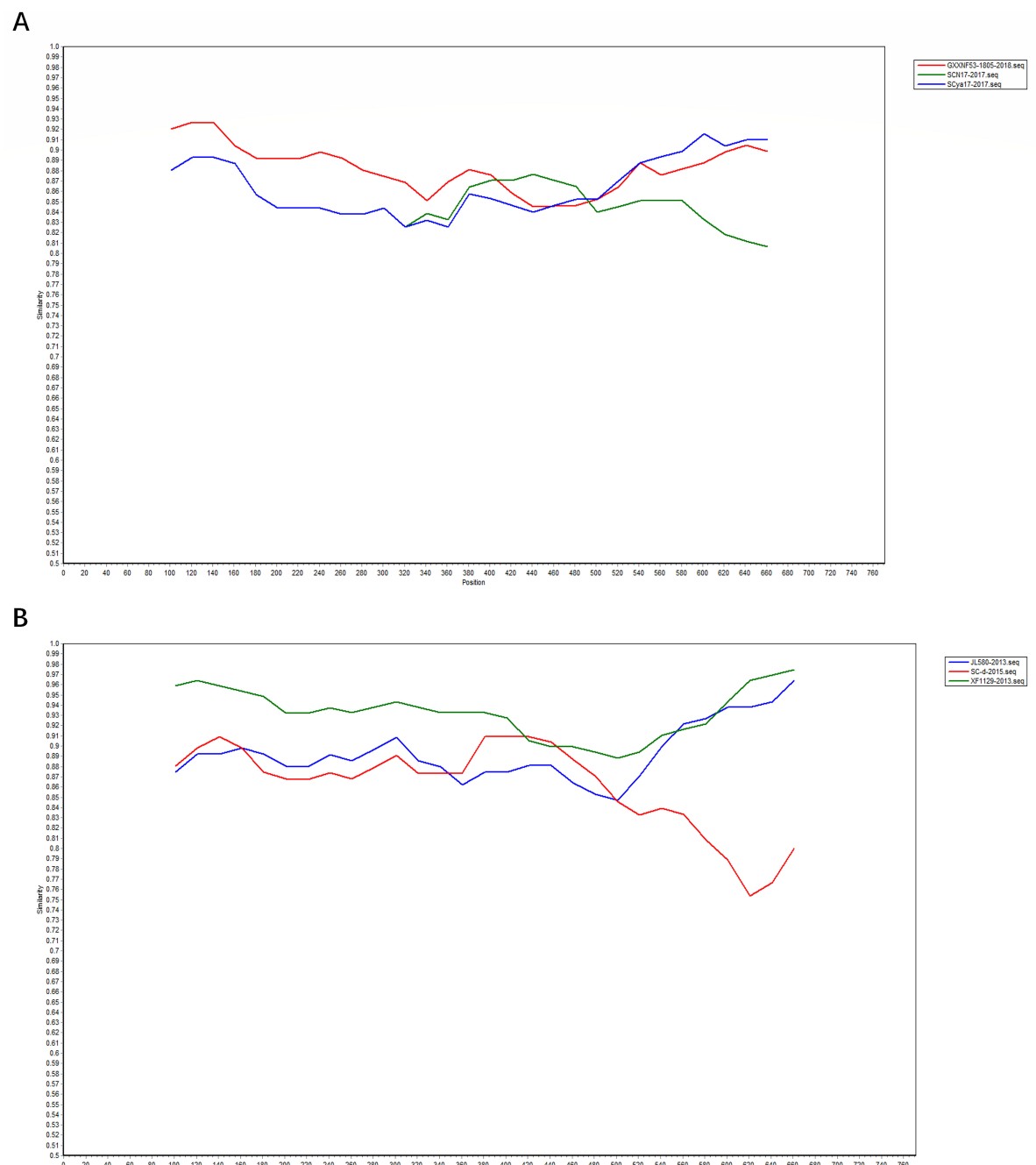

**FIG 6** Validation of *GP2* gene recombination events by SimPlot. The horizontal coordinate is position and the vertical coordinate is similarity. (A) Recombination validation results of the recombinant strain SCya17-2017. (B) Recombination validation results of the recombinant strain JL580-2013. When the dot plots of the reference sequences show nearly parallel lines, it indicates that the sequence of the destination sequence is highly similar to that of one of the reference sequences and there is no recombination signal. When the dot plots of the reference sequences show crosses, it indicates that a recombination signal may be present.

improve the situation that PRRSV is highly prone to recombination and mutation and generates new strains.

The study of the GP2 protein of PRRSV-2 strains prevalent in China from 1996 to 2023 shows that PRRSV-2 is in a state of persistent and widespread circulation and mutation. Recombination and amino acid comparison analyses of the GP2 protein made it clear that the most prevalent lineage of PRRSV-2 in China in recent years is lineage

1, which has shown a trend of continuous mutation and recombination, leading to a greater difference in the pathogenicity of different strains. This situation makes the prevention of the virus more difficult and reminds us that the development of vaccines should strengthen the prevention effect of lineage 1 strains. In 2006, the emergence of JXA1 as a representative of highly pathogenic strains increased the challenge of preventing PRRSV-2. The genetic characteristics as well as the pathogenicity of highly pathogenic strains have changed considerably. The continuous development of vaccines and drugs has not changed the fact that the virus continues to spread and recombine, and new mutated strains continue to emerge. More recombination and mutation of the GP2 protein of PRRSV-2 will also occur in the future. Accelerating and deepening the research on the genetic characterization and development of PRRSV-2 is the basis for improving the widespread transmission and pathogenicity of PRRSV-2, and the molecular mechanism of high mutation and recombination in the PRRSV genome still needs to be explored by more researchers.

## Conclusions

From 1996 to 2023, PRRSV-2 viruses prevalent in China mainly belong to lineages 1, 3, 5, and 8, with lineages 1 and 8 being the most prevalent PRRSV-2 strains in China. In terms of GP2 proteins, lineage 1 is the most recombinant lineage with the most frequent amino acid sequence substitutions, and the amino acid substitution site of lineage 8 is usually different from those of other lineages. This study provides scientific data for the genetic evolution and recombination analysis of GP2 in PRRSV-2, which will help to monitor the genetic variation of PRRSV and develop a safer and more efficient vaccine with broad-spectrum protection.

## AUTHOR AFFILIATIONS

[1]Guangdong Provincial Key Laboratory of Animal Molecular Design and Precise Breeding, School of Animal Science and Technology, Foshan University, Foshan, Guangdong, China
[2]College of Veterinary Medicine, Henan University of Animal Husbandry and Economy, Zhengzhou, Henan, China
[3]Gladstone Institutes of Virology and Immunology, University of California, San Francisco, California, USA

## AUTHOR ORCIDs

Mengmeng Zhao ⓘ http://orcid.org/0000-0003-4143-3606

## FUNDING

| Funder | Grant(s) | Author(s) |
| --- | --- | --- |
| Characteristic Innovation Project of Guangdong Provincial Department of Education | 2023KTSCX128 | Mengmeng Zhao |
| National Natural Science Foundation of China | 31902279 | Mengmeng Zhao |

## AUTHOR CONTRIBUTIONS

Kexin Liu, Conceptualization, Data curation, Formal analysis, Methodology, Software, Validation, Writing – original draft | Chen Lv, Conceptualization, Methodology, Validation, Writing – original draft | Cuihua He, Data curation, Formal analysis, Methodology | Jiankun Pang, Data curation, Formal analysis | Chunyao Lai, Project administration | Siliang Chen, Project administration | Ruining Wang, Supervision | Weili Kong, Supervision | Mengmeng Zhao, Funding acquisition, Investigation, Project administration, Supervision.

## ADDITIONAL FILES

The following material is available online.

## Open Peer Review

**PEER REVIEW HISTORY (review-history.pdf).** An accounting of the reviewer comments and feedback.

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
