## [Reviewer comments · Microbiology Spectrum]

Microbiology Spectrum

Analysis of the genetic evolution and recombination of the PRRSV-2 GP2 protein in China from 1996 to 2023

Kexin Liu, Chen Lv, Cuihua He, Jiankun Pang, Chunyao Lai, Siliang Chen, Ruining Wang, Weili KONG, Jun Ma, and Mengmeng Zhao

Corresponding Author(s): Mengmeng Zhao, Foshan University School of Life Science and Engineering

Review Timeline:

Submission Date:	November 26, 2024
Editorial Decision:	December 28, 2024
Revision Received:	February 4, 2025
Accepted:	February 17, 2025

Editor: Day-Yu Chao

Reviewer(s): Disclosure of reviewer identity is with reference to reviewer comments included in decision letter(s). The following individuals involved in review of your submission have agreed to reveal their identity: Jayeshbhai Mansinhbhai Chaudhari (Reviewer #1)

Transaction Report:

DOI: <https://doi.org/10.1128/spectrum.03079-24>

Re: Spectrum03079-24 (Analysis of the genetic evolution and recombination of the PRRSV-2 GP2 protein in China from 1996 to 2023)

Dear Dr. mengmeng Zhao:

Thank you for the privilege of reviewing your work. Below you will find my comments, instructions from the Spectrum editorial office, and the reviewer comments.

Revision Guidelines

Sincerely,
Day-Yu Chao
Editor
Microbiology Spectrum

Reviewer #1 (Comments for the Author):

Porcine reproductive and respiratory syndrome virus (PRRSV) continues to pose a significant threat to swine health worldwide. Its high genetic diversity and frequent recombination events make the development of a universal and effective vaccine particularly challenging. Consequently, it is essential to continually monitor and evaluate emerging recombinant variants to control the spread of the virus, primarily by restricting animal movement.

In this study, the authors analyzed the GP2 diversity of circulating PRRSV strains in China, focusing on lineages 1, 3, 5, and 8. GP2, a minor structural protein of PRRSV, plays a critical role in facilitating viral entry into host cells. Thus, tracking variants with altered GP2 sequences is of paramount importance.

The manuscript is well-written; however, I have the following concerns:

1. The introduction lacks sufficient detail about the significance of GP2 in viral infection. The authors should consider incorporating the following references in both the introduction and discussion sections.

I. A Single Amino Acid Substitution in Porcine Reproductive and Respiratory Syndrome Virus Glycoprotein 2 Significantly Impairs Its Infectivity in Macrophages. (doi: 10.3390/v14122822)

II. The Significance of the 98th Amino Acid in GP2a for Porcine Reproductive and Respiratory Syndrome Virus Adaptation in Marc-145 Cells (doi: 10.3390/v16050711)

2. Line 43: The correct term is Arteriviridae, not Arteritis virus.

3. Line 44: Please clarify the term "capsular membrane."

4. Lines 83-84: The significance of this sentence is unclear. Please elaborate or consider rephrasing.

5. Figures 1 and 3 are not cited in the text. Please ensure all figures are referenced appropriately.

6. Have the recombinant strains identified in this analysis been evaluated for differences in pathogenicity? Are there studies available that assess their altered pathogenic potential?

Reviewer #2 (Public repository details (Required)):

all the sequence data, if yet to public available, needs to be deposited

Reviewer #2 (Comments for the Author):

The manuscript presents a comprehensive analysis of the genetic evolution and recombination of the PRRSV-2 GP2 protein, highlighting its significance in understanding the epidemiology and informing vaccine development. This study is timely and relevant, given the ongoing challenges posed by PRRSV in swine production. However, several aspects of the manuscript could benefit from clarification, additional detail, and improved presentation to enhance its scientific rigor and impact.

Major Comments

1. While the study aims to fill the gap in research on the GP2 protein, the manuscript does not clearly articulate how the findings advance the understanding of PRRSV evolution compared to previous studies. Explicitly describe how the data on GP2 complements or extends prior findings on other structural proteins (e.g., GP5).

2. The manuscript highlights two recombination events but provides limited biological interpretation of their implications. Discuss the potential functional consequences of these recombination events on PRRSV fitness, pathogenicity, or immune escape.

3. The manuscript identifies lineage 1 as having higher recombination and mutation rates compared to lineage 8, which is described as more conserved. Elaborate on the potential evolutionary pressures driving these differences and their implications for vaccine design.

4. The manuscript mentions glycosylation of the GP2 protein but does not analyze its conservation across lineages. Given the known role of glycosylation in immune evasion, include an analysis of glycosylation sites and their variability in the dataset.

5. Figures illustrating recombination events (Figures 4 and 5) lack sufficient resolution and annotation to be fully interpretable. Provide clear labels, scales, and legends to enhance understanding. Include a summary table or heatmap comparing the mutation frequencies across lineages for a more accessible visualization of amino acid substitutions.

6. The phylogenetic trees (Figures 1 and 2) should be accompanied by a clear description of how bootstrap values support the clustering of lineages. Highlight key strains and their relevance to the evolutionary history of PRRSV.

Minor Comments

1. The introduction provides an adequate background on PRRSV but lacks focus on the biological significance of GP2. Expand on why GP2, compared to other structural proteins, warrants a detailed evolutionary analysis.

2. Provide more details on the selection criteria for the 64 representative strains. Explain how these strains were chosen to ensure broad lineage representation.

3. Ensure consistent use of terminology. For instance, "lineage" and "strain" are sometimes used interchangeably, which could confuse readers.

4. Clarify the statistical thresholds for significance in recombination analysis. Include a brief explanation of why specific recombination detection methods (e.g., RDP, GENECONV) were chosen.

5. The discussion could benefit from more integration with broader PRRSV literature. Compare findings on GP2 with genetic analyses of other PRRSV proteins to contextualize the results.

6. The manuscript contains grammatical errors and awkward phrasing that detract from clarity. A thorough language edit is recommended to improve readability.

The manuscript presents a comprehensive analysis of the genetic evolution and recombination of the PRRSV-2 GP2 protein, highlighting its significance in understanding the epidemiology and informing vaccine development. This study is timely and relevant, given the ongoing challenges posed by PRRSV in swine production. However, several aspects of the manuscript could benefit from clarification, additional detail, and improved presentation to enhance its scientific rigor and impact.

Major Comments

1. While the study aims to fill the gap in research on the GP2 protein, the manuscript does not clearly articulate how the findings advance the understanding of PRRSV evolution compared to previous studies. Explicitly describe how the data on GP2 complements or extends prior findings on other structural proteins (e.g., GP5).
2. The manuscript highlights two recombination events but provides limited biological interpretation of their implications. Discuss the potential functional consequences of these recombination events on PRRSV fitness, pathogenicity, or immune escape.
3. The manuscript identifies lineage 1 as having higher recombination and mutation rates compared to lineage 8, which is described as more conserved. Elaborate on the potential evolutionary pressures driving these differences and their implications for vaccine design.
4. The manuscript mentions glycosylation of the GP2 protein but does not analyze its conservation across lineages. Given the known role of glycosylation in immune evasion, include an analysis of glycosylation sites and their variability in the dataset.
5. Figures illustrating recombination events (Figures 4 and 5) lack sufficient resolution and annotation to be fully interpretable. Provide clear labels, scales, and legends to enhance understanding. Include a summary table or heatmap comparing the mutation frequencies across lineages for a more accessible visualization of amino acid substitutions.
6. The phylogenetic trees (Figures 1 and 2) should be accompanied by a clear description of how bootstrap values support the clustering of lineages. Highlight key strains and their relevance to the evolutionary history of PRRSV.

Minor Comments

Introduction:

The introduction provides an adequate background on PRRSV but lacks focus on the biological significance of GP2. Expand on why GP2, compared to other structural proteins, warrants a detailed evolutionary analysis.

Materials and Methods:

Provide more details on the selection criteria for the 64 representative strains. Explain how these strains were chosen to ensure broad lineage representation.

Terminology:

Ensure consistent use of terminology. For instance, "lineage" and "strain" are sometimes used interchangeably, which could confuse readers.

Statistical Analysis:

Clarify the statistical thresholds for significance in recombination analysis. Include a brief explanation of why specific recombination detection methods (e.g., RDP, GENECONV) were chosen.

Discussion:

The discussion could benefit from more integration with broader PRRSV literature. Compare findings on GP2 with genetic analyses of other PRRSV proteins to contextualize the results.

Language and Style:

The manuscript contains grammatical errors and awkward phrasing that detract from clarity. A thorough language edit is recommended to improve readability.

Conclusion and Recommendations

The manuscript addresses an important topic and provides valuable insights into PRRSV evolution and recombination. However, addressing the above issues will significantly enhance the manuscript's scientific impact and clarity. I recommend major revisions to address these concerns before publication.

Dear Reviewer 1:

Thank you for your constructive comments. We have made some changes to the manuscript. These changes will not influence the content and framework of the manuscript. Here we list the changes and marked them in yellow in the revised manuscript. We have also responded to each of your comments and suggestions in a point-by-point manner. We appreciate your warm work earnestly and hope that the correction will meet with approval.

1. The introduction lacks sufficient detail about the significance of GP2 in viral infection. The authors should consider incorporating the following references in both the introduction and discussion sections.

I. A Single Amino Acid Substitution in Porcine Reproductive and Respiratory Syndrome Virus Glycoprotein 2 Significantly Impairs Its Infectivity in Macrophages. (doi: 10.3390/v14122822)

II. The Significance of the 98th Amino Acid in GP2a for Porcine Reproductive and Respiratory Syndrome Virus Adaptation in Marc-145 Cells (doi: 10.3390/v16050711)

Response: We have added them to lines 231-237 of the revised manuscript. These two references are numbered 40 and 41, respectively.

Lines 231-237: The T98→S98 mutation decreases the infectivity of PRRSV-2 toward Marc-145 cells [40]. This mutation is known to occur in lineages 1 and 3 based on amino acid sequence comparison, which suggests that lineages 1 and 3 may be less effective at isolating the virulent strain when Marc-145 cells are used. According to the results of the amino acid sequence comparison, no amino acid substitution occurred at K160, and the degree of conservation was extremely high. When K160 was replaced, the infectivity of PRRSV to PAMs was significantly reduced, and the replication of Low-Passage PRRSV Strain in pigs was reduced [41], which can be applied to the experimental production of vaccines to improve vaccine safety.

2. Line 43: The correct term is Arteriviridae, not Arteritis virus.

Response: Thank you for pointing out the mistake, we have revised it in line 43.

3. Line 44: Please clarify the term "capsular membrane."

Response: Capsular membrane refers to the lipid-like bilayer of proteins, polysaccharides, and lipids encapsulated in the viral capsid.

4. Lines 83-84: The significance of this sentence is unclear. Please elaborate or consider rephrasing.

Response: We have removed it.

5. Figures 1 and 3 are not cited in the text. Please ensure all figures are referenced appropriately.

Response: Thank you for your suggestion. However, Figure 1 is recited in line 129, and Figure 3 is referenced in line 143.

6. Have the recombinant strains identified in this analysis been evaluated for differences in pathogenicity? Are there studies available that assess their altered pathogenic potential?

Response: We did not conduct studies on the pathogenicity of these two recombinant strains. According to our search, we know that JL580 has been researched to classify as a recombinant highly pathogenic strain. Investigations have shown that between 2016 and 2017, clinical

samples of sick piglets, including lungs and lymph nodes, were collected from two different pig farms in Sichuan province, southwestern China, and that affected pigs had high fevers (40.2-42.1 °C) and showed clear signs of respiratory disease, with an incidence rate of 45.3% for the SCya17 strain and a mortality rate of 4.8%.

Reference: Whole Genome Analysis of Two Novel Type 2 Porcine Reproductive and Respiratory Syndrome Viruses with Complex Genome Recombination between Lineage 8, 3, and 1 Strains Identified in Southwestern China. *Viruses*. (doi: 10.3390/v10060328.)

Regards!

Mengmeng Zhao

Dear Reviewer 2:

Thank you for your constructive comments. We have made some changes to the manuscript. These changes will not influence the content and framework of the manuscript. Here we list the changes and marked them in yellow in the revised manuscript. We have also responded to each of your comments and suggestions in a point-by-point manner. We appreciate your warm work earnestly and hope that the correction will meet with approval.

Major Comments

1. While the study aims to fill the gap in research on the GP2 protein, the manuscript does not clearly articulate how the findings advance the understanding of PRRSV evolution compared to previous studies. Explicitly describe how the data on GP2 complements or extends prior findings on other structural proteins (e.g., GP5).

Response: Thank you for your suggestion. We have added lines 243-247 to summarize how the findings advance the understanding of PRRSV evolution. Lines 200-202, 204-207, and 216-219 compare GP2 with GP3, GP5, and NSP2 to complement the prior findings.

Lines 243-247: Recombination and amino acid comparison analyses of the GP2 protein made it clear that the most prevalent lineage of PRRSV-2 in China in recent years is lineage 1, which has shown a trend of continuous mutation and recombination, leading to a greater difference in the pathogenicity of different strains. This situation makes the prevention of the virus more difficult and reminds us that the development of vaccines should strengthen the prevention effect of lineage 1 strains.

Lines 200-202: The minimum values of nucleotide and amino acid similarity of GP3 and GP5 are lower than the minimum values of nucleotide and amino acid similarity of GP2 [35, 36]. Compared with GP3 and GP5, which are also structural proteins, GP2 is more conserved and less heterogeneous.

Lines 204-207: For example, in China, recombination events of NSP2, the non-structural protein of PRRSV-2, occurred mainly in lineage 8 [37]. However, according to the recombination and amino acid mutation of GP2 in this study, lineage 1 is more consistent with the characteristics of high recombination frequency and multiple mutations.

Lines 216-219: Compared to the GP5 protein of lineage 8, the GP5 protein was recombined at a higher frequency with greater differences in similarity. In connection with the idea that GP2 proteins influence the invasion of virulent strains into host cells, the high conservation of lineage 8 in the GP2 may correlate with the high pathogenicity of lineage 8 strains [38].

2. The manuscript highlights two recombination events but provides limited biological interpretation of their implications. Discuss the potential functional consequences of these recombination events on PRRSV fitness, pathogenicity, or immune escape.

Response: We have discussed the potential functional consequences of these recombination events on PRRSV in lines 191-194.

Lines 191-194: SCya17 and JL580, as recombinant strains, differed significantly in pathogenicity, with JL580 being a highly pathogenic strain and SCya17 showing low pathogenicity in clinical investigations [33]. Recombination has resulted in significant genetic diversity of NADC30-like PRRSV in China [34], which is one of the reasons for the large differences in viral pathogenicity.

3. The manuscript identifies lineage 1 as having higher recombination and mutation rates compared to lineage 8, which is described as more conserved. Elaborate on the potential evolutionary pressures driving these differences and their implications for vaccine design.

Response: We have illustrated the potential evolutionary pressures driving these differences and their implications for vaccine design in lines 212-216.

Lines 212-216: Lineage 8 tends to be conservative, indicating that this lineage is not as popular as lineage 1 in China in recent years, and the strains of lineage 1 have better epidemic conditions in China. Conventional vaccines for PRRSV in the Chinese market are mainly developed with lineage 8 strains, such as CH-1R, GDr180, JXA1-R, TJM-F92, etc. As a result, the cross-protection of conventional vaccines against lineage 1 is weak, which makes lineage 1 more susceptible to widespread epidemics and recombination. Therefore, vaccine development should consider strengthening the prevention against the more prevalent lineage strains in recent years.

4. The manuscript mentions glycosylation of the GP2 protein but does not analyze its conservation across lineages. Given the known role of glycosylation in immune evasion, include an analysis of glycosylation sites and their variability in the dataset.

Response: We have analyzed the conservation versus variability of glycosylation sites across lineages in lines 229-231.

Lines 229-231: The glycosylation sites of GP2 are 178 and 184, and glycosylation at position 184 has been shown to be essential for PRRSV transmission [39]. Amino acid sequence analysis showed no mutations in either glycosylation site, indicating that the glycosylation sites used for sequence analysis of the strains were highly conserved.

5. Figures illustrating recombination events (Figures 4 and 5) lack sufficient resolution and annotation to be fully interpretable. Provide clear labels, scales, and legends to enhance understanding. Include a summary table or heatmap comparing the mutation frequencies across lineages for a more accessible visualization of amino acid substitutions.

Response: We have provided clearer images and added more annotations to make it easier for readers to understand. We have produced a heat map of the lineage with its amino acid mutation sites in line 151.

6. The phylogenetic trees (Figures 1 and 2) should be accompanied by a clear description of how bootstrap values support the clustering of lineages. Highlight key strains and their relevance to the evolutionary history of PRRSV.

Response: Thank you for your suggestion. We have visualized bootstrap values between 0.7 and 1 in Figures 1 and 2 to facilitate the observation of correlations between key strains and PRRSV evolutionary history.

Minor Comments

1. The introduction provides an adequate background on PRRSV but lacks focus on the biological significance of GP2. Expand on why GP2, compared to other structural proteins, warrants a detailed evolutionary analysis.

Response: Thank you for your suggestion. We have elucidated why a detailed evolutionary analysis of the GP2 protein is needed in lines 61-64.

Lines 61-64: In summary, the GP2 protein plays a key role in viral invasion, replication, and apoptosis and is closely related to the specific antibodies produced by the host immune response. Therefore, a detailed evolutionary analysis of GP2 would be beneficial to discover

ways to effectively block the invasion of PRRSV into host cells.

2. Provide more details on the selection criteria for the 64 representative strains. Explain how these strains were chosen to ensure broad lineage representation.

Response: We have provided more details on the selection criteria for the 64 representative strains in lines 83-87.

Lines 83-87: Among these amino acid sequences, 64 PRRSV GP2 sequences were selected (Table 1). The criteria for sequence selection included classical strains, strains often used as lineage representatives in the literature, vaccine strains commonly used in China, and newly emerged strains in recent years. The classical strains included VR2332, RespRRS MLV, and NADC34 from the USA, and CH-1a and BJ-4 from China. Lineage representative strains include HENAN-XINX (lineage 1), JL580 (lineage 1), QYYZ (lineage 3), FJFS (lineage 3), S1 (lineage 5), GS2003 (lineage 5), JXA1 (lineage 8), HUN4 (lineage 8), and so on.

3. Ensure consistent use of terminology. For instance, "lineage" and "strain" are sometimes used interchangeably, which could confuse readers.

Response: We have checked the full text to ensure consistency of terminology. However, lineage refers to the internal delineation of genotypes, and strains usually refer to the individual PRRSVs isolated, and it is hard to express both in a single word.

4. Clarify the statistical thresholds for significance in recombination analysis. Include a brief explanation of why specific recombination detection methods (e.g., RDP, GENECONV) were chosen.

Response: We have clarified the statistical thresholds of significance for recombination analyses in line 180 of the revised manuscript, and the reasons for choosing specific recombination detection methods have been stated in lines 98-100.

Line 180: The p -value for both reorganization events was less than 0.05, which is statistically significant.

Lines 98-100: The RDP software was widely used for the detection of recombination events [31], allowing highly automated analysis of a large number of sequences with multiple methods [32].

5. The discussion could benefit from more integration with broader PRRSV literature. Compare findings on GP2 with genetic analyses of other PRRSV proteins to contextualize the results.

Response: Lines 200-202, 204-207, and 216-219 have comparisons of the genetic analysis of GP2 with the genetic analysis of other PRRSV proteins.

Lines 200-202: The minimum values of nucleotide and amino acid similarity of GP3 and GP5 are lower than the minimum values of nucleotide and amino acid similarity of GP2 [35, 36]. Compared with GP3 and GP5, which are also structural proteins, GP2 is more conserved and less heterogeneous.

Lines 204-207: For example, in China, recombination events of NSP2, the non-structural protein of PRRSV-2, occurred mainly in lineage 8 [37]. However, according to the recombination and amino acid mutation of GP2 in this study, lineage 1 is more consistent with the characteristics of high recombination frequency and multiple mutations.

Lines 216-219: Compared to the GP5 protein of lineage 8, the GP5 protein was recombined at a higher frequency with greater differences in similarity. In connection with the idea that GP2 proteins influence the invasion of virulent strains into host cells, the high

conservation of lineage 8 in the GP2 may correlate with the high pathogenicity of lineage 8 strains [38].

6. The manuscript contains grammatical errors and awkward phrasing that detract from clarity. A thorough language edit is recommended to improve readability.

Response: Thank you for the suggestion. We have carefully reviewed the manuscript to ensure that the English grammar and writing format meet the required standards.

Regards!

Mengmeng Zhao

Re: Spectrum03079-24R1 (Analysis of the genetic evolution and recombination of the PRRSV-2 GP2 protein in China from 1996 to 2023)

Dear Dr. Mengmeng Zhao:

Your manuscript has been accepted, and I am forwarding it to the ASM production staff for publication. Your paper will first be checked to make sure all elements meet the technical requirements. ASM staff will contact you if anything needs to be revised before copyediting and production can begin. Otherwise, you will be notified when your proofs are ready to be viewed.

Sincerely,
Day-Yu Chao
Editor
Microbiology Spectrum

Reviewer #1 (Comments for the Author):

The authors have addressed all comments, and I have no further feedback.

Reviewer #2 (Comments for the Author):

This manuscript presents a detailed genetic and recombination analysis of the PRRSV-2 GP2 protein using 570 nucleotide sequences collected over a 27-year period (1996-2023) in China. The study provides insights into phylogenetics, lineage-specific mutations, recombination events, and potential implications for vaccine development. Given the persistence and genetic diversity of PRRSV, this research is highly relevant to understanding the virus's evolutionary dynamics, which is critical for improving disease control and vaccine efficacy.